# Real-time ultrafast oscilloscope with a relativistic electron bunch train

In Hyung Baek [1,6], Hyun Woo Kim [1,6], Hyeon Sang Bark[1,6], Kyu-Ha Jang[1], Sunjeong Park[1], Junho Shin[1], Young Chan Kim[1], Mihye Kim[1], Key Young Oang[1], Kitae Lee[1,2], Fabian Rotermund [3], Nikolay A. Vinokurov [4,5] & Young Uk Jeong [1,2✉]

The deflection of charged particles is an intuitive way to visualize an electromagnetic oscillation of coherent light. Here, we present a real-time ultrafast oscilloscope for time-frozen visualization of a terahertz (THz) optical wave by probing light-driven motion of relativistic electrons. We found the unique condition of subwavelength metal slit waveguide for preserving the distortion-free optical waveform during its propagation. Momentary stamping of the wave, transversely travelling inside a metal slit, on an ultrashort wide electron bunch enables the single-shot recording of an ultrafast optical waveform. As a proof-of-concept experiment, we successfully demonstrated to capture the entire field oscillation of a THz pulse with a sampling rate of 75.7 TS/s. Owing to the use of transversely-wide and longitudinally-short electron bunch and transversely travelling wave, the proposed "single-shot oscilloscope" will open up new avenue for developing the real-time petahertz (PHz) metrology.

---

[1] Korea Atomic Energy Research Institute (KAERI), Daejeon 34057, Republic of Korea. [2] University of Science and Technology (UST), Daejeon 34113, Republic of Korea. [3] Korea Advanced Institute of Science and Technology (KAIST), Daejeon 34141, Republic of Korea. [4] Budker Institute of Nuclear Physics SB RAS, Novosibirsk 630090, Russia. [5] Novosibirsk State University, Novosibirsk 630090, Russia. [6]These authors contributed equally: In Hyung Baek, Hyun Woo Kim, Hyeon Sang Bark. ✉email: yujung@kaeri.re.kr

Real-time measurement of an ultrafast waveform has been a constant desire in many fields of fundamental science and technology[1]. Although extreme ultraviolet pulses opened new frontiers for PHz optical metrology[2–4], offering a single-shot measurement is still challenging. Recently, temporal imaging with a time lens has emerged as a method for single-shot acquisition of optical waveform[5–7], but its temporal accuracy should be further improved for broader use. Also, in such nonlinear optical (NLO) conversion techniques for assessing broadband light waves[8], the reconstructed field distribution is prone to be distorted from the original waveform due to imperfect phase-matching features inside NLO materials during frequency conversion processes[9–11]. In this context, a charged particle is considered as a most reliable probe of electromagnetic waves because its deflection motion in free space can directly reflect the spatiotemporal field distribution of optical waves without requiring the use of any NLO parametric media. Recently, direct visualization of ultrafast light oscillation was demonstrated by tracing the kinetic energy change of electrons released from molecules[12], metal tips[13,14], and photocathodes[15,16]. However, for the reconstruction of an overall field trace, data should be collected by scanning the relative time delay of probe electrons because the size of electrons sources is not sufficiently large to visualize an entire waveform.

Here, we take a fresh approach to the real-time oscilloscope via momentary stamping of a traveling optical wave on a quasi-one-dimensional (Q-1D) array (i.e., short, vertically thin, and horizontally wide bunch) of relativistic (MeV) electrons whose velocity is close to the speed of light. Relativistic electrons are definitely beneficial for ultrashort bunch generation because their temporal broadening induced by the space charge effect can be significantly suppressed in the relativistic regime[17,18]. For the temporal characterization of relativistic electron bunches, electromagnetic waves have extensively contributed in two diagnostic ways: electro-optic detection with near-infrared laser pulses[19] and streaking with a radio frequency (RF) wave[20] or a terahertz (THz) wave[21–23]. In this study, we propose a concept that overturns the conventional streaking technique, i.e., measuring the instant longitudinal dependence of the electric field of a light wave by using an ultrashort relativistic electron bunch, and present a proof-of-concept experiment with optical wave packets oscillating at THz frequency.

## Results

**Operation principle of real-time ultrafast oscilloscope.** The operation principle of our real-time ultrafast oscilloscope is shown in Fig. 1a. As an electron source, we used a laser-driven RF photocathode[18,24–26] because it can generate relativistic electron bunches synchronized with an optical wave providing an ultrashort bunch duration, and low emittance and high brightness. Two subwavelength metal slits are placed along the path of the ultrashort electron bunches. The first thick slit trims the transverse beam shape from circular to line shaped and entirely blocks the residual portion of the circular electron beam. An unknown optical signal with a linear electric polarization ($\overrightarrow{E_0}$) is coupled laterally into the gap of the second thin metal slit. For a spectrally

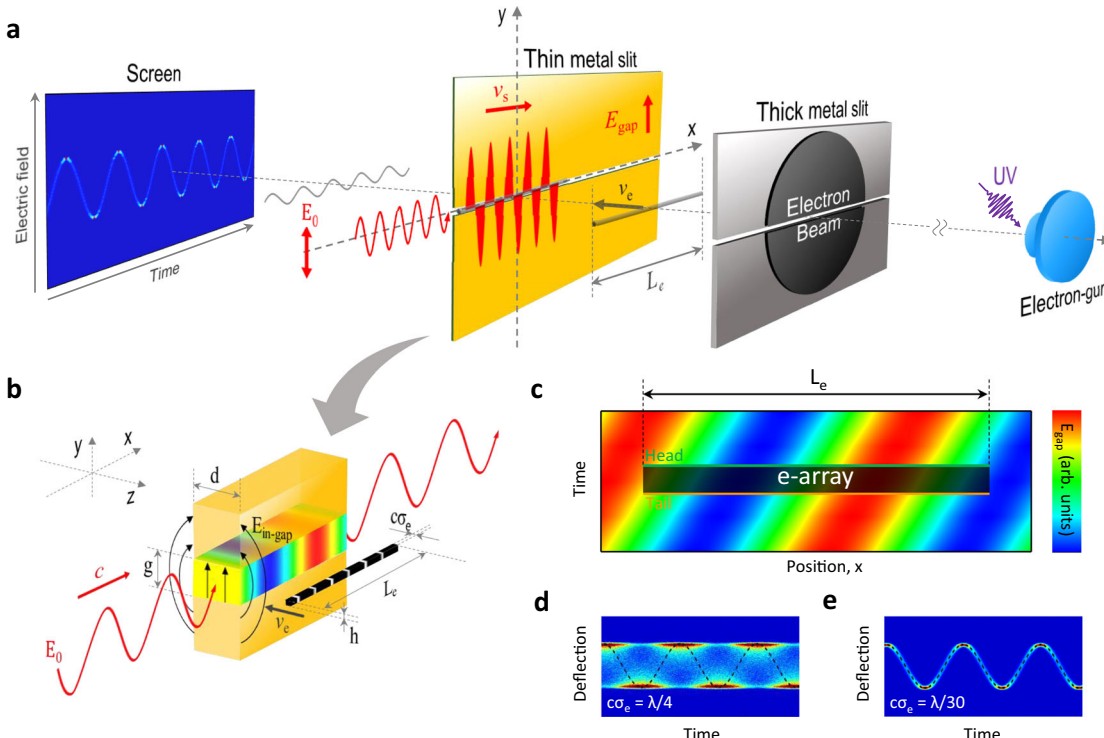

**Fig. 1 Conceptual illustration of the real-time ultrafast oscilloscope. a** Schematic setup for the real-time measurement of an unknown optical signal by using a MeV electron bunch. The ultrashort electron bunch is generated from a UV-induced electron gun, and its velocity ($\boldsymbol{v_e}$) is close to the speed of light. The input optical signal with linear electric polarization is coupled laterally into the gap of a thin metal slit and propagates along the gap. The Q-1D electron array is deflected by the amplified electric field inside the metal gap and then recorded on the screen. Here, the time window for single-shot acquisition depends only on the horizontal width ($L_e$) of the electron beam. **b** Magnified illustration of the orthogonal interaction between the optical signal and Q-1D electron array with a longitudinal length $c\sigma_e$ and a horizontal width $L_e$, where $c$ is the speed of light and $\sigma_e$ is the electron bunch duration. **c** Electric field mapping inside the gap depending on the time and x-position. The black rectangle depicts the $y$-projection of the Q-1D electron array in (**b**). Green and orange solid lines indicate the first and last rows of the Q-1D electron array. **d, e** Superimposition effect of waveforms depending on the relationship between $\sigma_e$ and $\lambda$ calculated for intuitive comparison of two extreme conditions: $\boldsymbol{c\sigma_e = \lambda/4}$ (**d**) and $\boldsymbol{c\sigma_e = \lambda/30}$ (**e**).

broad signal, the excessive pulse broadening owing to group velocity dispersion does not occur for the TEM mode of the thin parallel-plate metal waveguide[27]. The confined electric field at the gap (in-gap field, $E_{in-gap}$) is enhanced by the subwavelength size of the gap. Being thin in $y$-direction and wide in $x$-direction, electron beam may be represented as an array of many thin round beams with different $x$-coordinates, or "columns". Electrons of each such "column" experiences a different electric field value when it encounters the propagating wave inside the gap. Its deflection angle along the $y$-axis is determined by the integrated electric field along the $z$-axis within an effective thickness of the slit. As a result, the optical waveform of an unknown signal can be directly reconstructed on a screen by merging individual deflections of all electron "columns". Note that the magnetic field component of the incident optical wave does not affect the motion of electrons because its magnetic polarization is parallel to the propagation direction of the electron beam.

Figure 1b shows the orthogonal electron-wave interaction to describe the signal integrity of our electron oscilloscope. The electric field enhancement factor inside the gap is determined by geometric parameters of the metal structure, such as the gap size ($g$) and the slit thickness ($d$), for a given wavelength ($\lambda$) of the incident wave. Although the Q-1D electron array feels the longitudinally integrated field within the effective length of field which can be expressed as $L_{eff} = \frac{g}{2} \ln \frac{8\lambda}{\pi g |\zeta|}$, where $\zeta$ is the surface impedance of metal slit, while it propagates between both metal plates, no deformation of the waveform reconstructed by all electron columns occurs within the characteristic length ($x_{max} < g/|\zeta|$) which means a distance from the

entrance of waveguide, if $L_{eff}$ is sufficiently shorter than $\lambda$ (see Supplementary Note 1 for details). However, the rms length of the electron bunch ($c\sigma_e$) induces the waveform distortion in this oscilloscope as follows. The continuous phase shift of the deflection field within the electron bunch duration causes temporal blurring of the waveform, as shown in Fig. 1c–e (see "Simulation" in "Methods" for details). Hence the time dependence of deflection field in the gap (Fig. 1c) leads to waveform distortion from a sinusoidal to square-like wave due to the horizontal superimposition effect of the traveling wave on the screen (Fig. 1d, e). In addition, the emittance of electron beam can cause blurring of the waveform if the beam pointing spread is larger than a single-pixel size in a CCD image.

**Visualization of single-shot THz waveform**. To demonstrate our idea experimentally, we utilized a relativistic electron accelerator and a THz wave as electron and light sources, respectively (see "Methods" and ref. [28] for details). A 1 mm-thick tantalum (Ta) slit with a gap of 30 μm is used for tailoring the transverse shape of the electron beam. The quasi-single-cycle THz pulse with a maximum field strength of 215 kV/cm is temporally synchronized with the electron bunch and then focused on a side of a copper (Cu) slit with a thickness of 25 μm and a gap of 30 μm (Fig. 2a). The Q-1D electron array, whose charge is estimated to be about 10 fC, encounters the enhanced THz in-gap field. Subsequently, electrons carrying the THz waveform are detected by a p43 phosphor screen with an electron-photon conversion efficiency of 200 and an electron-multiplying CCD (EMCCD) camera. Optical delay tuning of the incident pulse provides real-time (50 Hz in this experiment) panning

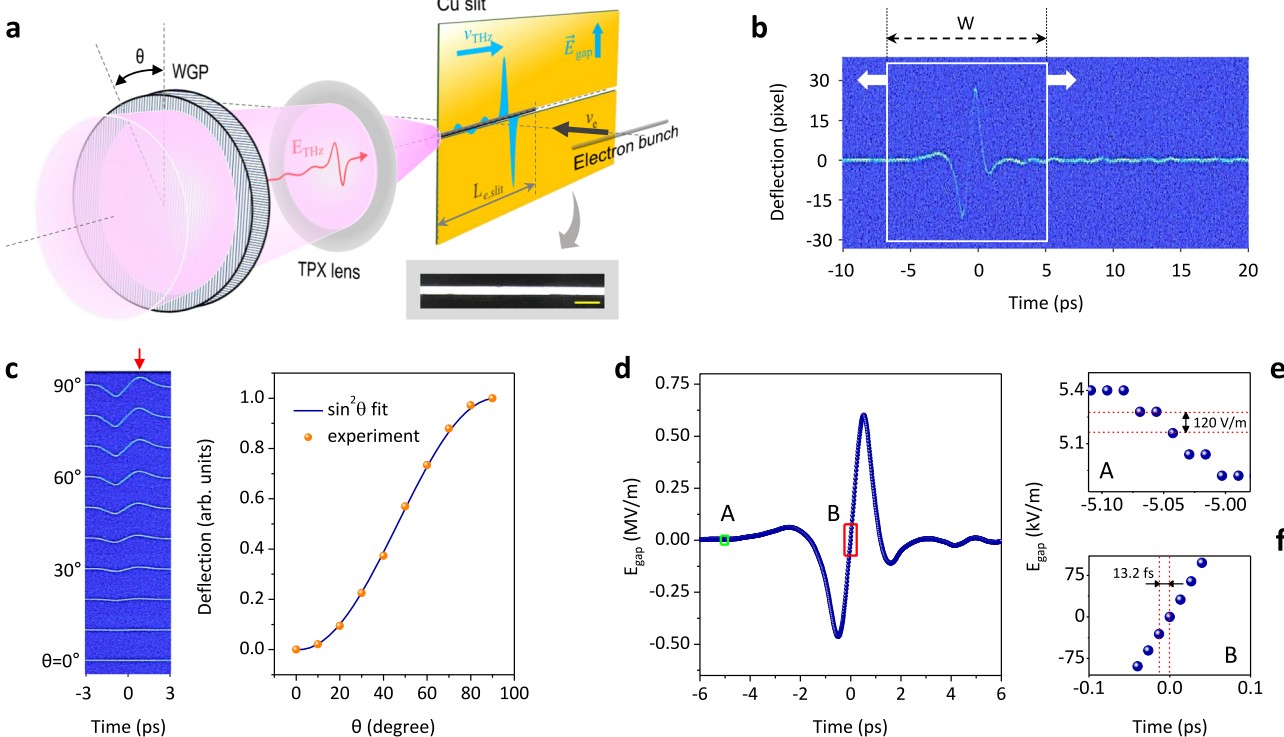

**Fig. 2 Proof-of-concept experiments for THz optical waveform measurement. a** Simple layout of the experimental setup. As the incident optical signal for the proof-of-concept experiment, a vertically polarized THz wave was focused on the thin copper slit by a single spherical lens. A wire-grid polarizer pair was utilized to control the electric field strength of the input THz signal. The fabricated copper slit with a gap size of 30 μm is shown in the inset image with a scale bar of 100 μm. **b** THz waveform measured by the EMCCD. The white box represents a single-shot time window ($W$) for $L = 3.78$ mm at the gap. By adjusting the optical delay between the input signal and the electron bunch, the region of interest in the time domain can be easily changed. **c** The experimental results on the detection linearity of our oscilloscope. The positive maximum values of the deflected electron array in the left image are plotted in the graph as a function of the relative angle ($\theta$) of the wire-grid polarizer pair (orange dots). The blue solid curve is the quadratic sinusoidal fit to the experimental data. **d** Time-varying $E_{in-gap}$ from the single-shot data in the white box of (**b**). **e, f** Magnified plots of green and red boxes in (**d**) for presenting the accuracy of the vertical field amplitude (**e**) and temporal step per pixel of EMCCD camera (**f**).

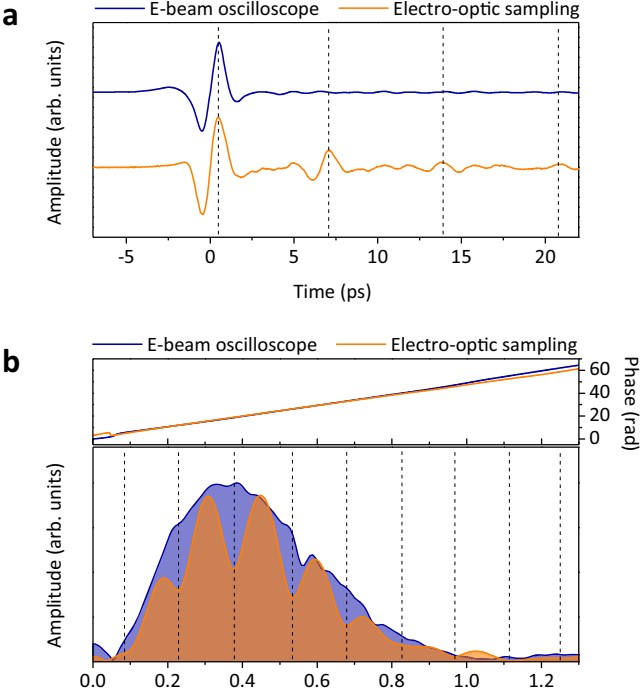

**Fig. 3 Waveform comparison with conventional EOS. a** Single-shot THz waveform recorded by the electron beam (E-beam) oscilloscope, and time trace of the THz field obtained by the EOS technique with an optical delay scan. The vertical dashed lines indicate the positions of the main and echo signals. **b** Fourier-transformed spectra of THz signals in the time domain (**a**), and their phase information. The amplitudes of both spectra are normalized for clarity of comparison. The vertical dashed lines indicate the periodicity of the spectral modulation.

of a single-shot time window ($W$) of 12.6 ps over the entire waveform, as shown in Fig. 2b. By extracting the waveform from the CCD image (Fig. 2d), this oscilloscope yields a THz field oscillogram with a vertical field accuracy of 120 V/m and a temporal step per pixel ($R$) of 13.2 fs, as shown in Fig. 2e, f (see Supplementary Note 2 for details). For the recorded waveform with a $W$ of 12.6 ps, one pixel for the single temporal step is comprised of about 66 electrons via phosphor screen. Additionally, the detection linearity was proven by measuring THz waveforms by varying the angle of a wire-grid polarizer from 0 to 90°. Figure 2c shows the positive peak values of the deflection strength in ten oscillograms are well fitted by the square of a sine function.

To further evaluate our experimental results, we compared THz waveforms measured by both the conventional electro-optic sampling (EOS) technique and our real-time electron oscilloscope under the same experimental environment (see "Methods" for details). The electro-optically recorded waveform contains echoes of the main signal with a time interval of 6.78 ± 0.21 ps (Fig. 3a) coming from the multiple reflections at the boundary between air and the GaP crystal. These echoes shorten the available time window for the Fourier transform and lead to spectral modulation with a period of 147.5 ± 6.34 GHz (Fig. 3b). In contrast, the relativistic electron oscilloscope shows a clear distinction in the signal integrity because the direct electron-field interaction in vacuum is inherently free from echoes, bandwidth limitations of the electro-optic crystal[29] and complex distortions that typically originate from nonlinearities of the optical medium[30]. As we expected from Fig. 1e, the THz waveform measured by the relativistic electron oscilloscope is

absolutely distortion-free due to the extremely high $\lambda/c\sigma_e$ of ~108. Moreover, together with our simulation results on $d$ and $c\sigma_e$ for a given $\lambda$, the almost analogous phase evolutions, central frequencies of ~0.37 THz, and spectral bandwidths of ~0.44 THz in both spectra guarantee the fidelity of our stamping technique.

**Real-time control of signal sampling rate.** The convenient zoom-in and -out of the measured waveform are fascinating functions of this electron oscilloscope because R depends on the electron beam divergence. Figure 4a illustrates the principle of time-base control and waveforms recorded on the EMCCD for three different divergences of the electron beam. The horizontally over-focused electron beam enables the oscilloscope to obtain a higher temporal accuracy per pixel of the EMCCD by a factor of $1 + (2D\tan\theta/L_{e,slit})$ compared to the case of the collimated electron beam, where $D$ is the distance from the slit to the screen and $\theta$ is the divergence half-angle of the electron beam. A quadrupole magnet is employed for controlling $\theta$ and the corresponding $L_{e,slit}$ to control both $R$ and $W$, as shown in Fig. 4b, c. We observed that both $R$ and $W$ increase with the horizontal focusing strength ($k$). For configuration III ($k = 89\,\mathrm{m}^{-2}$) in Fig. 4a, our oscilloscope visualizes the THz waveform with $R$ of 0.565 fs in $W$ of 273 fs, corresponding to a sampling rate of 1.77 PS/s in real time.

## Discussion

Although the bandwidth of this oscilloscope is estimated to be ~1 THz due to a limited bunch duration of our electron source and an insufficient effective length of field, our simulation provides a possibility to improve the overall performance for detecting an ultrafast waveform (see Supplementary Note 4 in details). Superconducting waveguide can be a simple way to increase the characteristic length for measuring a waveform with higher frequency. We believe that technical advances[31–33] in attosecond electron bunch generation and ultra-sensitive electron detector[34] can overcome the current limitation in the near future.

In conclusion, we devised a real-time ultrafast oscilloscope based on an ultrashort electron bunch train and successfully demonstrated single-shot measurement of a THz wave with a frame rate of 50 Hz as a proof-of-concept experiment. The direct imprinting of an in-gap field preserving the incident waveform on a moving electron beam allows us to observe a distortion-free full-field of light wave. Since the demonstrated concept can be further available for the characterization of optical signals with any frequency or longitudinal shape, future works will be focused on the experimental demonstration at higher frequency, for instance, in the near- and mid-infrared frequency regions. We expect that our study will be the first step towards real-time PHz oscilloscope technology.

## Methods

**Experimental setup.** We used a 1 kHz Ti:sapphire regenerative amplifier (Spitfire Ace-35F1K, Spectra Physics) delivering 35 fs (FWHM) laser pulses at a wavelength of 800 nm. The laser beam with an average power of 5 W was divided by a 9:1 beam splitter to generate a THz pulse and an electron bunch. From a Cu cathode, the initial electron bunch with a charge of 1 pC was generated by irradiating the third harmonic radiation (150 μJ per pulse) of the fundamental laser pulse and then accelerated up to 3.1 MeV by a 1.5 cell RF cavity fed by an S-band (2.856 GHz) klystron at a repetition rate of 50 Hz. The duration of the ultraviolet pulse was measured to be 130 fs (FWHM) by characterizing the second-harmonic radiation in the self-conjugating optical cross-correlator. The electron bunch was compressed down to 25 fs (rms) using its energy chirp via isochronous bend with two 45° magnets at the metal slit in ultrahigh vacuum (~10⁻⁸ Torr). The bunch duration of the relativistic electron was measured by a THz streak camera[25]. The normalized emittance of the electron beam was measured to be 0.25 mm mrad in both axes. The deflected electrons generated photons at the phosphor screen. These photons were reflected by a flat gold mirror with an incident angle of 45° and then detected by an EMCCD (Andor iXon Ultra 888, Oxford Instruments) with a 60 mm F2.8D microlens.

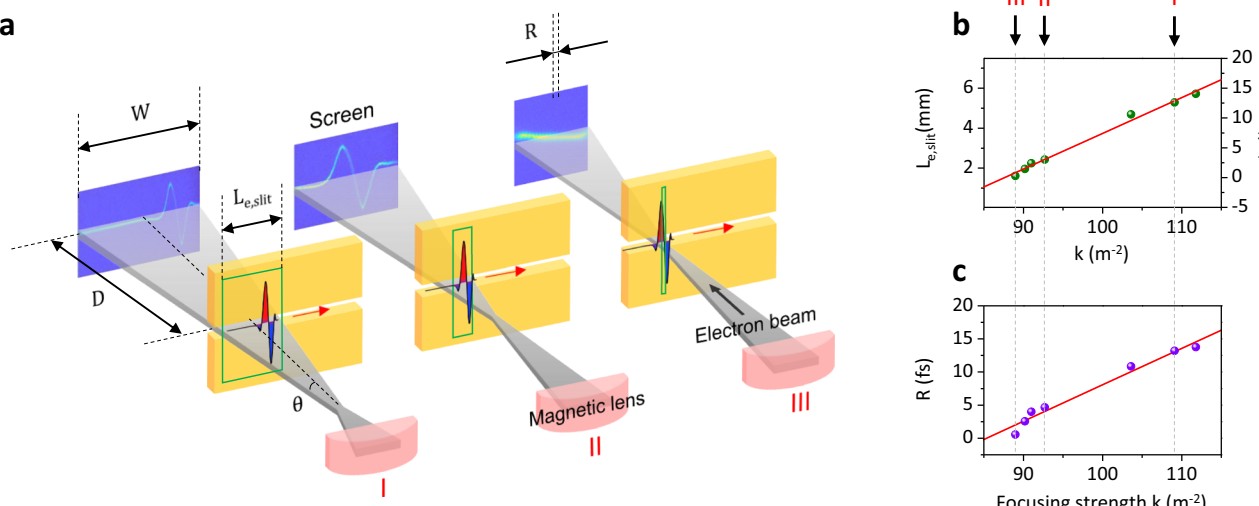

**Fig. 4 Feasibility of real-time PHz oscilloscope. a** Controlling the single-shot time window ($W$) on the screen and the temporal resolution per single pixel ($R$) of our oscilloscope. The three green boxes indicate different interaction regions of the Q-1D electron array and input optical signal at the Cu slit, where $L_{e,gap}$ is the horizontal length of the electron beam passing through the gap. The focusing strength ($k$) of the electron beam is controlled by the applied current of the quadrupole magnet. **b, c** Measured data (dots) and linear fits (solid curves) of $L_{e,gap}$, $W$, and $R$ as a function of $k$. The black arrows correspond to the three experimental images in (**a**).

**Simulations**. When the synchronized ultrashort electron bunch passes through the slit, it obtains vertical momentum, which is induced by the incident optical field. This in-gap field, $E_{in\text{-}gap}(t)$, is given by the following equation:

$$E_{in\text{-}gap}(t) = E_p \sin(\omega t)e^{-(t/\tau_{laser})^2} \qquad (1)$$

where $E_p$ is the peak electric field, $\omega = 2\pi c/\lambda$ is the angular frequency, $c$ is the speed of light, $\lambda$ is the wavelength of the incident field, and $\tau_{laser}$ is the duration of the laser pulse. The gain of the electron vertical momentum is $\Delta p_y(t) = eL_{eff}E_{in\text{-}gap}(t_e)/c$, where $L_{eff}$ is the effective length of the induced electric field inside the slit and $t_e$ is the moment of time when electron passes the middle of the slit. We assumed that the electron bunch has a longitudinal Gaussian distribution. The initial vertical beam divergence ($y'_{ini}$) of the electron beam and the kinetic energy $p$ of the electron bunch in the propagation direction are set to 40 μrad and 3.1 MeV, respectively. The vertical divergence angle of electrons after the slit is $y' = y'_{ini} + \Delta p_y/p$.

**Slit fabrication**. The metal slit structure was fabricated by the careful alignment of two 25 μm-thick Cu foils with a purity of 99.8% (46365, Alfa Aesar) in parallel. This Cu slit was mechanically mounted by polyethylene plates for electric isolation.

**THz pulse generation and characterization**. The quasi-single-cycle THz pulse was generated by optical rectification in a 1.3 mol% MgO-doped stoichiometric LiNbO₃ crystal. A diffraction grating with a groove density of 1800 mm⁻¹ and a single spherical lens ($f$ = 150 mm) were used for tilting the pulse front of the 800 nm pulse. The average power of the 50 Hz THz pulse train was measured to be 83.5 μW by a calibrated pyroelectric detector (THZ5B-MT-DZ, Gentec-EO) at the slit position. The THz beam was focused laterally into the gap of the slit by a TPX lens with a focal length of 50 mm. Its focal spot diameter was 1 mm (FWHM), as measured by a pyroelectric CCD camera (IR/V-T0831, NEC Corp.). The maximum THz electric field strength of the 215 kV/cm was analytically estimated (see Supplementary Note 3 for details). The THz time trace and its corresponding spectrum were obtained by EOS with a 0.3 mm-thick GaP crystal located at the same position as the Cu slit for a reliable comparison with the electron oscilloscope.

**Data analysis**. All waveforms were imprinted on a single electron bunch, so the relative timing jitter between the optical wave and the electron bunch at the slit can be considered negligible for distortion. The scales for both the time and field strength ($E_{in\text{-}gap}$) on the EMCCD were calibrated by moving the high-accuracy optical delay stage (M-ILS200HA, Newport Corporation) and by using the formula $E_{in\text{-}gap} = pc\Delta y/(eDL_{eff})$ where $\Delta y$ is the amount of $y$-axis deflection, and $p$ is the momentum of electron, $L_{eff} = \int_{-\infty}^{\infty} E_y(x, 0, z, t)dz/E_y(x, 0, 0, t)pc\Delta y/(eDL_{eff})$ is calculated effective length of the field. Each dot composing the THz waveform in Fig. 2d is the central value, which is obtained by a Gaussian fit along the deflection

axis in the CCD image of Fig. 2b. For extraction of the single-shot time window size and the temporal resolution per pixel in all measured images, a pixel pitch of 32 μm was used, where the focal length of the light-collection lens was 60 mm.

## Data availability
The data that support the findings of this study are available from the corresponding authors upon reasonable request.

## Code availability
The codes and simulation files that support the plots and data analysis within this paper are available from the corresponding author upon reasonable request.

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

## Acknowledgements

This work was supported by the World Class Institute (WCI) Program of the National Research Foundation of Korea (NRF), funded by the Ministry of Science, ICT and Future Planning (NRF grant no. WCI 2011-001), and an internal R&D programme at KAERI funded by the Ministry of Science and ICT (MSIT) of the Republic of Korea (524450-20), and a National Research Council of Science & Technology (NST) grant by the Korea government (MSIT) (no. CAP-18-05-KAERI).

## Author contributions

I.H.B., H.W.K., S.P., K.Y.O., N.A.V. and Y.U.J. carried out the real-time electron oscilloscope experiment. I.H.B., Y.C.K., M.K., K.L. and F.R. led the generation and characterization of THz field. J.S. contributed to the temporal stability improvement of electron bunch train. H.W.K., H.S.B., K-H.J. and K. Y. O. carried out the simulations and data analysis. I.H.B. and Y.U.J. wrote the manuscript, with input from all authors.

## Competing interests

The authors declare no competing interests.
