## [Peer Review File. · Nature Communications]

REVIEWER COMMENTS

Reviewer #1 (Remarks to the Author):

This work is heavily based on previous work by various groups, and unfortunately, these work are not included in the references. For instance, in "Electron microscopy of electromagnetic waveforms, Science, 353, 374 (2016)" the basic concept of using electron beam to sample the electromagnetic waveforms has been proposed and demonstrated. In this manuscript, a particular metallic structure that helps to preserve the waveform is used, but the concept is similar. This manuscript combines two techniques to measure the waveform of a THz pulse: one is that the waveform of a THz pulse may be preserved in a parallel-plate metal waveguide, and the other is electron beam may be deflected by the THz field. Both techniques have been demonstrated by various groups and combining these techniques for measuring THz pulse shape is straightforward and lacks the novelty to warrant publication in Nat. Comm.

Regarding the significance of this technique to the field, I would say the significance is low. Using a relativistic electron beam to measure THz waveform is certainly not a generic method. As is well-known, accelerator is complicated and expensive, I can't imagine who will build an accelerator in order to measure THz waveform. There are so many easier methods in optics community for this purpose. Even with such a complicated setup, the authors only reported measurement of a typical THz pulse that was also measured by EOS technique. I expect at least some measurements that can't be done with EOS should be reported in this manuscript. However, only a very ordinary measurement is reported.

It is not clear to what extent the THz pulse waveform is preserved in the metal waveguide. I'd like to point out that what's measured is the integrated transverse kick from Lorentz force during the passage through the metal waveguide, not the instantaneous THz field in the slit, not the original THz field either. The metal waveguide might enhance some of the frequency component of the THz pulse while suppress the other, leading to distortions in the measurement, and more detailed discussion and experimental verification is required to support the conclusion.

In summary, I can't recommend its publication in Nat. Comm, because it clearly doesn't meet the standards for innovation and impact. This work lacks a clear motivation and using a relativistic electron beam to sample the THz field limits its application to special cases.

Reviewer #3 (Remarks to the Author):

Ultrafast electron diffraction (UED) is a well-established technique of material structure study invoking laser pulse as a pump and an electron pulse as a probe. Authors of the current work are utilizing relativistic electrons (~ 3 MeV) together with achromatic bend, which improves one of the main difficulties related to the technique, i.e., electron bunch dispersion and broadening, giving at the final state the electron bunch duration of 25 fs. By very slight modification of the UED setup well-described in [1], Baek I.H. et al. turned their facility into an ultrafast oscilloscope with ability to record the electromagnetic waveforms in a single-shot regime and with implemented control of oscilloscope's time base. By choosing a decent time window, the timing jitter between the electron bunch and electromagnetic wave can be excluded.

The presented results are convincing enough for the proof-of-principle experiment and are of good quality. At the current stage, the limit of the technique in spectral range is determined mainly by the duration of the electron bunch and is ≤ 10 THz. Although, when the problem might be solved by invoking mid/near-infrared or visible fields for the electron bunch compression to a femto-/sub-femtosecond level, as authors have mentioned, the reviewer sees some problems with space charge due to longitudinal or transversal focusing (despite of self-focusing) would come into play and would strongly limit the signal and resolution at shorter wavelengths of waveforms.

While the work is lacking great novelty, at a quick glance this experimental technique has vast application area and, as it is, could already find its usage in observing and studying, e.g., THz (spoof) surface plasmons [2] with a possibility of applying velocity matching or electron bunch front tilting [3], thus, supporting other methods in this field of research. I would recommend the work for publishing in

Nature Communications.

Reviewer's comments/recommendations:

1. The claim of absence of group velocity dispersion in the waveguide is valid for broadband THz pulses (<10 THz), however, for ultrabroadband THz or, e.g., MIR pulses it is, generally speaking, not true.
2. Although, it is rather clear what variables v_s, E_I in the Fig.1a stand for, but they are not defined either in text/supplementary or figure caption; E_0 is defined in supplementary.
3. A value of the electron bunch charge at the sample's vicinity could be included. The reviewer could estimate it below 10 fC from the previous work [1] but it is for slightly different setup configuration. The stated sampling rate of the oscilloscope will make more sense with known number of electrons in the burst.
4. The work could have included estimation of electromagnetic waveform resolution limits with dependences on a) electron bunch duration; b) electron bunch charge incorporating beam emittance. In the reviewer's opinion, such estimation would better give feeling of the technique capability and would show improvement targets without lowering the impact of the work.

Response to reviewer's comments

We would like to thank all reviewers for their efforts and constructive criticism, which allowed us to further deepen the understanding of our studies and improve the quality of our manuscript. We have taken most of the reviewer's comments to heart and revised the manuscript to clarify all the points. To each reviewer comment written in boldface, we have tried to give our response (in blue color) in detail.

● Comments of the reviewer #1:

1. This work is heavily based on previous work by various groups, and unfortunately, these work are not included in the references. For instance, in “Electron microscopy of electromagnetic waveforms, Science, 353, 374 (2016)” the basic concept of using electron beam to sample the electromagnetic waveforms has been proposed and demonstrated. In this manuscript, a particular metallic structure that helps to preserve the waveform is used, but the concept is similar. This manuscript combines two techniques to measure the waveform of a THz pulse: one is that the waveform of a THz pulse may be preserved in a parallel-plate metal waveguide, and the other is electron beam may be deflected by the THz field. Both techniques have been demonstrated by various groups and combing these techniques for measuring THz pulse shape is straightforward and lacks the novelty to warrant publication in Nat. Comm.

⇒ We regret to disagree with the reviewer's opinion. Although we added the recommended reference (“Electron microscopy of electromagnetic waveforms” in Science) in text as #16, the referred paper and our work share a sampling tool only (electron beam) to detect a waveform. Basic concept is little bit similar but operation mechanism is definitely distinguished each other. Our oscilloscope demonstrates the direct visualization of single-shot ultrafast waveform without any optical or electrical parametric process for the first time to our knowledge. The “Key word” of our work is the single-shot electron stamping of optical waveform in real-time by utilizing a particular metallic structure as the reviewer mentioned above. The unique preservation of waveform acts a core role for single-shot oscilloscope. This is based on a much simpler way than any other techniques. Furthermore, it is more powerful and accurate for transferring an initial waveform. We believe that this new straightforward technique has a general impact for the oscilloscope technology and in accordance with the scope of *Nature Communications*.

2. Regarding the significance of this technique to the field, I would say the significance is low. Using a relativistic electron beam to measure THz waveform is certainly not a generic method. As is well-known, accelerator is complicated and expensive, I can't imagine who will build an accelerator in order to measure THz waveform. There are so many easier methods in optics community for this purpose. Even with such a complicated setup, the authors only reported measurement of a typical THz pulse that was also measured by EOS technique. I expect at least some measurements that can't be done with EOS should be reported in this manuscript. However, only a very ordinary measurement is reported.

⇒ We disagree with the reviewer's point that the electron beam technology is less accessible. High-energy e-beam may require facility-scale accelerator, but, low-energy e-beam with a few MeV or less is available at laboratory scale. As shown in our previous papers, we have demonstrated that bright and femtosecond electron beam generation is possible with an electron gun and a simple optics. Contrary to the reviewer's point, we expect that low-energy electron beam with a pulse duration of a few fs or less will be quickly available in many laboratories just by using small-scale and inexpensive devices.

Compared to this technique, the electron microscopy is much more complex and expensive. And as we described in introduction of main text, there are many methods in optics, but they cannot be free from signal distortion. Our technique guarantees the distortion-free waveform within the specific range (characteristic length). This work is focused on the visualization of single-shot waveform and its reliability. As the essential purpose of oscilloscope is a visualization of waveform as it is. This ‘very ordinary measurement’ is exactly what we want to report by elucidating its essential function as a new oscilloscope.

3. It is not clear to what extent the THz pulse waveform is preserved in the metal waveguide. I’d like to point out that what’s measured is the integrated transverse kick from Lorentz force during the passage through the metal waveguide, not the instantaneous THz field in the slit, not the original THz field either.

⇒ Thanks for the appropriate comment. We added to the "Section 1 in Supplementary Information" the estimate of "effective length" for the transverse kick (Eq. (7) & Eq. (8)). The decrease of this effective length at the frequency growth also limits the bandwidth. The conclusion after Eq. (8) is "Therefore, one can say that the bandwidth of the ‘oscilloscope’ is about 1 THz."

4. The metal waveguide might enhance some of the frequency component of the THz pulse while suppress the other, leading to distortions in the measurement, and more detailed discussion and experimental verification is required to support the conclusion.

⇒ We also added to the "Section 1 in Supplementary Information" the estimate of dispersion and attenuation of the waveguide (Eq. (1) to (6)). The conclusion after them is "For 10 THz and copper slit waveguide, $x_{max} < 300g \approx 10$ mm. It means that at shorter distances than 10 mm (from the waveguide entrance) there is no distortion for signals with a bandwidth of 10 THz in our waveguide."

Page 3 in main text: “Although the Q-1D electron array feels the longitudinally integrated field within the effective length of field which can be expressed as $L_{eff} = \frac{g}{2} \ln \frac{8\lambda}{\pi g |\zeta|}$, where ζ is the surface impedance of metal slit, while it propagates between both metal plates, no deformation of the waveform reconstructed by all electron columns occurs within the characteristic length ($x_{max} < g/|\zeta|$) which means a distance from the entrance of waveguide, if L_{eff} is sufficiently shorter than λ (see Supplementary Section 1 for details).”

In summary, I can’t recommend its publication in Nat. Comm, because it clearly doesn’t meet the standards for innovation and impact. This work lacks a clear motivation and using a relativistic electron beam to sample the THz field limits its application to special cases.

⇒ We strongly point out that the rationale for the reviewer’s negative judgment is not logically appropriate. As mentioned above, the inaccessibility of the ultrashort electron beam is somewhat exaggerated. We added in Supplementary Information that the wave distortion and streaking resolution of the slit waveguide are technical issues, not fundamental limitations. Our results show that shorter electron bunches and new slit structure can measure higher frequency electromagnetic wave in real time without distortion.

Section 1 in Supplementary Information:

“Section 1. Calculation of the effective length of field

Electromagnetic wave in the gap of the slit is the TEM one, see Fig. S1.

Figure S1. The scheme of experiment. The 1D electron array meets the electric field (TEM mode) confined at gap.

It is easy to understand, as there is a two-wire line of two strips, and the gap g between strips is much less than the wavelength λ divided by 2π . The propagation direction of wave is x , therefore $E_x = H_x = 0$. Transverse components of electric field can be found from solution of 2-D electrostatic problem. For simple analytic estimate we will consider thick slit with a of gap g , $2\pi d/g \gg 1$ (see Figure S2).

Figure S2. Electric field at the edge of thick slit. The 1D electron array meets the electric field (TEM mode) confined at gap.

Then the corresponding potential $\varphi = \varphi_0 \text{Im}\Psi/\pi$ can be easily found using conformal mapping $z +$

$iy = g(s - \ln \sqrt{s+1} + \ln \sqrt{s-1})/\pi$, $s = \sqrt{e^\Psi + 1}$, where φ_0 is the potential of the top electrode. Then

$$E_y + iE_z = -\frac{\varphi_0}{\pi} \frac{d\psi}{d(z+iy)} = -\frac{2\varphi_0}{g} \frac{1}{s} = \frac{E_{max}}{s}, \quad (1)$$

For $|s| > 1$ $s \approx \sqrt{\frac{\pi^2(z+iy)^2}{g^2} + 2}$ and

$$E_y + iE_z \approx \frac{g}{\pi} \frac{E_{max}}{\sqrt{(z+iy)^2 + 2g^2/\pi^2}} \quad (2)$$

Due to finite conductivity of the slit metal, which is described by Leontovich boundary conditions with the surface impedance ζ , the phase velocity of this wave is slightly lower than the speed of light in vacuum. It can be estimated using common formula for the attenuation in the waveguide

$$Imk_x = \frac{Re\zeta}{2} \int |H|^2 dx / \int |H|^2 dy dz \approx \frac{Re\zeta}{g}, \quad (3)$$

In the first approximation, k_x has to be linear function of the surface impedance, therefore

$$k_x \approx k + \zeta \frac{iImk_x}{Re\zeta} \approx k + i \frac{\zeta}{g}. \quad (4)$$

For metals with conductivity σ it gives

$$k_x \approx k + \frac{1+i}{g} \sqrt{\frac{ck}{8\pi\sigma}}. \quad (5)$$

Then one of the limitations for the bandwidth of such oscilloscope can be expressed as

$$|k_x - k|x_{max} \approx |\zeta|x_{max}/g < 1, \quad (6)$$

where x_{max} is the distance from the slit entrance. For 10 THz and copper slit $|\zeta|$ is approximately 3×10^{-3} . Therefore, $x_{max} < 300g \approx 10$ mm. It means that at shorter distances than 10 mm there is no distortion for signals with a bandwidth of 10 THz in our waveguide. In our experiment, the electron beam is incident at a point within 5 mm from the entrance of the slit. Therefore, the wave read by the electron beam is in a state without distortion for frequency up to a few tens THz.

Let electron coordinates are $[x, 0, v(t - t_1)]$. Then the transverse momentum variation of electron passed through the slit is

$$\begin{aligned}
\Delta p_y &= e \int_{-\infty}^{\infty} E_y(x, 0, vt - vt_1, t) dt \\
&\cong \frac{e}{v} 2E_{max} \cos \left[\omega \left(t_1 - \frac{x}{c} \right) \right] \int_{-d/2}^{\infty} \frac{\cos \left[\omega \left(z + \frac{d}{2} \right) / v \right]}{s} dz \\
&\cong \frac{e}{v} 2E_{max} \cos \left[\omega \left(t_1 - \frac{x}{c} \right) \right] \left[\left(\frac{2v}{\omega} - g \right) \sin \left(\frac{\omega d}{2v} \right) + \frac{2g}{\pi} \cos \left(\frac{\omega d}{v} \right) K_0 \left(\frac{\omega g}{\pi v} \sqrt{2} \right) \right] \\
&\approx \frac{e}{v} 2E_{max} \cos \left[\omega \left(t_1 - \frac{x}{c} \right) \right] \left[d + \frac{2g}{\pi} \left(\ln \frac{\pi v \sqrt{2}}{\omega g} - C \right) - \frac{\omega g d}{2v} \right] \\
&= \frac{e E_{max} L_{eff}}{c} \cos \left[\omega \left(t_1 - \frac{x}{c} \right) \right], \tag{7}
\end{aligned}$$

where, K_0 is the modified Bessel function and L_{eff} is the effective length of the electric field in z-axis at the gap center of the slit waveguide.

$$L_{eff} \approx d + \frac{2g}{\pi} \left(\ln \frac{\lambda}{g\sqrt{2}} - C \right) - \frac{\lambda g d}{\lambda}, \tag{8}$$

and $C \approx 0.577$ is Euler's constant. For $d = 25 \mu\text{m}$, $g = 30 \mu\text{m}$ and $\lambda = 0.6 \text{ mm}$, Eq. (8) gives $L_{eff} = 60 \mu\text{m}$. For $\lambda = 0.15 \text{ mm}$, $L_{eff} = 35 \mu\text{m}$. Therefore, one can say that the bandwidth of the "oscilloscope" is about 1 THz.

The corresponding deflection y on the screen of the EMCCD is given by expression

$$y = \frac{\Delta p_y}{p} D = \frac{e D L_{eff}}{pc} E_0 \left(t_1 - \frac{x}{c} \right), \tag{9}$$

where p is electron momentum, and D is the distance from the slit to the screen. Due to finite bunch duration σ_e , the line width in x coordinate is $c\sigma_e$. For $\sigma_e = 25 \text{ fs}$, it is $7.5 \mu\text{m}$, which corresponds to 2 pixels of the CCD."

● **Comments of the reviewer #3:**

Ultrafast electron diffraction (UED) is a well-established technique of material structure study invoking laser pulse as a pump and an electron pulse as a probe. Authors of the current work are utilizing relativistic electrons (~3 MeV) together with achromatic bend, which improves one of the main difficulties related to the technique, i.e., electron bunch dispersion and broadening, giving at the final state the electron bunch duration of 25 fs. By very slight modification of the UED setup well-described in [1], Baek I.H. et al. turned their facility into an ultrafast oscilloscope with ability to record the electromagnetic waveforms in a single-shot regime and with implemented control of oscilloscope's time base. By choosing a decent time window, the timing jitter between the electron bunch and electromagnetic wave can be excluded.

⇒ We appreciate the referee for careful reading our manuscript and acknowledging the novelty of our work.

The presented results are convincing enough for the proof-of-principle experiment and are of good quality. At the current stage, the limit of the technique in spectral range is determined mainly by the duration of the electron bunch and is $\lesssim 10$ THz. Although, when the problem might be solved by invoking mid/near-infrared or visible fields for the electron bunch compression to a femto-/sub-femtosecond level, as authors have mentioned, the reviewer sees some problems with space charge due to longitudinal or transversal focusing (despite of self-focusing) would come into play and would strongly limit the signal and resolution at shorter wavelengths of waveforms.

⇒ We appreciate the reviewer's comments. We fully agree with the referee's opinion that the space charge effect of extremely short electron bunch can induce disrupt the operation bandwidth of this oscilloscope. In this work, we used the pair of phosphor screen and EMCCD to record the trajectory of MeV electrons in order to avoid the damage of CCD cells. If the electron bunch with an extremely low charge and direct electron camera with an ultrahigh sensitivity in added reference #32 "A direct electron detector for time-resolved MeV electron microscopy in *Rev. Sci. Instrum.* **88**, 033702 (2017)" are utilized, we expect that the problem caused by the space-charge effect can be further alleviated.

While the work is lacking great novelty, at a quick glance this experimental technique has vast application area and, as it is, could already find its usage in observing and studying, e.g., THz (spoof) surface plasmons [2] with a possibility of applying velocity matching or electron bunch front tilting [3], thus, supporting other methods in this field of research. I would recommend the work for publishing in *Nature Communications*.

1. The claim of absence of group velocity dispersion in the waveguide is valid for broadband THz pulses (<10 THz), however, for ultrabroadband THz or, e.g., MIR pulses it is, generally speaking, not true.

⇒ We fully agree with the reviewer's opinion for a long propagation of light inside the waveguide. In our work, a finite characteristic length (x_{max}), which there is no distortion for recorded signals, As shown in added Eq. (4) and below of Supplementary Information, we can guarantee the maintenance of field shape within ~3 cm (corresponding to the single-shot time window of 100 ps, which is enough to visualize ultrashort electromagnetic signals) from the entrance of slit under our experimental conditions. For higher or broader frequency, this characteristic length, x_{max} becomes shorter. The simplest way

to increase the length (frequency) is to reduce the resistance by lowering the temperature of the metal slit. Additionally, it may be possible to use superconducting or photonic-crystal waveguides.

2. Although, it is rather clear what variables v_s , E_I in the Fig.1a stand for, but they are not defined either in text/supplementary or figure caption; E_0 is defined in supplementary.

⇒ We missed their definitions in text. The variable v_s and E_I in the Fig. 1b mean the velocity and the electric field amplitude of optical signal in free space, respectively. In fact, we were going to express that the variable E_0 in Fig. 1a indicates the electric field amplitude of linearly polarized waves. To avoid confusions, we replaced the v_s to c and added the definition of E_0 in text as below. And also we unified these variables to E_0 in all texts and figures.

Page 3 in main text: “An unknown optical signal with a linear electric polarization (\vec{E}_0) is coupled laterally into the gap of the second thin metal slit.”

3. A value of the electron bunch charge at the sample’s vicinity could be included. The reviewer could estimate it below 10 fC from the previous work [1] but it is for slightly different setup configuration. The stated sampling rate of the oscilloscope will make more sense with known number of electrons in the burst.

⇒ The reviewer’s estimation is quite exact. The charge of electron beam at the slit position could be calculated to be ~10 fC, when the diameter of electron beam with a charge of 1 pC at slit and the gap-size are considered. As the reviewer mentioned, this value is slightly different with that in our previous work because the oscilloscope operates as the over-focusing mode. As we mentioned in the previous answer, the waveform was recorded in CCD pixels through the electron-photon conversion at phosphor screen. The number of electrons in the single burst and the quantum efficiency of p43 phosphor screen are about 62,414 and 200, respectively. Hence, approximately 66 electrons (corresponding to ~13138 photons) make a single time-step in oscillogram. We added some sentences in the revised manuscript as below.

Page 4 in main text: “The Q-1D electron array, whose charge is estimated to be about 10 fC, encounters the enhanced THz in-gap field. Subsequently, electrons carrying the THz waveform are detected by a p43 phosphor screen with an electron-photon conversion efficiency of 200 and an electron-multiplying CCD (EMCCD) camera.”

Page 4 in main text: “For the recorded waveform with W of 12.6 ps, one pixel for the single temporal step is comprised of about 66 electrons via phosphor screen.”

4. The work could have included estimation of electromagnetic waveform resolution limits with dependences on a) electron bunch duration; b) electron bunch charge incorporating beam emittance. In the reviewer’s opinion, such estimation would better give feeling of the technique capability and would show improvement targets without lowering the impact of the work.

⇒ We really appreciate the reviewer for the fruitful suggestions. In fact, we performed the simulation about a THz waveform distortion depending on the beam emittance ($\varepsilon_{N,x,y}$) as shown Fig. S4 and confirmed it is well matched with experimentally recorded signals. The beam emittance-induced distortion of recorded waveform can be clearly seen, however, it is not easy to define the minimum value of $\varepsilon_{N,x,y}$ for getting a distortion-free signal. However, our simulation (Fig. S5) shows that the electron bunch duration (σ_e) of 667 attoseconds and $\varepsilon_{N,x,y}$ of 90 pm should be required to extend this

technique to NIR range at least. We added these estimations to the “middle of Page 6 in revised manuscript” and the “Section 4 in Supplementary Information” as below.

Page 6 in main text: “Although the bandwidth of this oscilloscope is estimated to be ~1 THz due to a limited bunch duration of our electron source and an insufficient effective length of field, our simulation provides a possibility to improve the overall performance for detecting an ultrafast waveform (see Supplementary Section 4 in details). Superconducting waveguide can be a simple way to increase the characteristic length for measuring a waveform with higher frequency. We believe that technical advances²⁹⁻³¹ in attosecond electron bunch generation and ultra-sensitive electron detector³² can overcome the current limitation in the near future.”

Section 4 in Supplementary Information: “We carried out the simulation about electron beam emittance-induced blurring effects. All parameters except an electron beam divergence are same with the simulation in Methods section which describes the concept of this work. The initial horizontal divergence of electron beam is estimated as $x'_{ini} = \varepsilon_{N,x}/\beta\gamma x_{ini}$, where $\varepsilon_{N,x}$ is the normalized horizontal beam emittance, β is the relativistic velocity, γ is the Lorentz factor and x_{ini} is the horizontal electron beam size (rms). The initial vertical divergence also can be defined as $y'_{ini} = \varepsilon_{N,y}/\beta\gamma y_{ini}$. Figure S4 shows the comparison of simulated THz waveforms whose beam emittances are applied or not. The THz waveform with a central frequency of 0.37 THz, a pulse duration of 2 ps, and an electric field strength of 25 MV/m was used as an input signal. The simulated waveform in Fig. S4c is well matched with our experimental data (Fig. 2b) in main text.”

Figure S4. THz waveform simulations depending on the electron beam emittance. a. Ideal case for $\varepsilon_{N,x,y} = 0$, b. Only vertical emittance ($\varepsilon_{N,y} = 0.3 \mu\text{m}$) is considered. c. The identical emittance ($\varepsilon_{N,x,y} = 0.3 \mu\text{m}$) for both horizontal and vertical axes is applied for simulations.

“To propose the applicability of our oscilloscope for extending its operation range, we performed a further simulation at NIR region. The peak electric field strength inside the slit is 150 MV/m, the central wavelength (λ) of input pulse is 800 nm, and the pulse duration is 5 fs (rms). Here, we changed the bunch length of 105 electrons and the normalized horizontal electron beam emittance at the fixed vertical one of 5.5 nm because the emittance effect at time-axis is to be seen clearly. For the electron bunch length of $\lambda/15$, the profile of waveform can be discerned even though $\epsilon_{N,x}$ is close to 90 pm which is corresponding to the horizontal divergence of 0.32 μ rad. However, for the electron bunch length of $\lambda/4$, $\epsilon_{N,x}$ of 30 pm at least should be required to visualize a contour of waveform as shown in Fig. S5b.”

Figure S5. NIR waveform simulations depending on the bunch length and emittance of electron beam.

REVIEWERS' COMMENTS

Reviewer #1 (Remarks to the Author):

The manuscript has been improved by adding supplementary materials to show the process of THz-electron interaction. My concern in the technical part has been largely addressed, but my concern on novelty and impact of this work, remains.

THz-electron interaction in a narrow slit has been experimentally studied in PRX 8, 021061 (2018) and PRAB 22, 012803 (2019) (should have been cited). Because the deflection of the electron depends on the phase of the interaction, one can either extract the information of the electron beam or the information of the THz pulse from measurement of the deflected beam distribution. The original goal of the scheme is to use the THz pulse to measure the electron pulse structure. Roughly speaking, a small accelerator including laser, rf source, electron gun, low-level RF system, magnets, power supply, etc. costs > \$ 1 M, but the THz source costs < \$ 0.1 M. So it is a small investment to build a THz source compared to the accelerator, but it allows the measurement of important information of the electron beam not accessible by other methods. The authors of the current manuscript wants to go the other way around, e.g. using the electron beam to measure THz pulse. The investment is high, and the gain is low. As shown in the paper, EOS can do it for most of the cases, I just can't imagine who will build a small accelerator to produce electron beam for this purpose unless they already have one. In Fig.3 the authors show the distortion from EOS by the echo signal. However, as long as one know it is an echo signal, one can remove it before FFT, which removes also most of the distortions in spectrum. EOS is a standard technique in THz spectroscopy which is widely used to study ultrafast dynamics. So the distortion is exaggerated in Fig.3. I have many friends working on THz spectroscopy, and I don't hear anyone complaining about the echo signal.

In the reply, the authors said, 'We strongly point out that the rationale for the reviewer's negative judgment is not logically appropriate. As mentioned above, the inaccessibility of the ultrashort electron beam is somewhat exaggerated.' I don't agree with this statement. There is no doubt that there are >10 labs in universities that have small accelerators to produce relativistic electron beam. But the question is how many of them can produce <100 fs beam and how many of them have the need to measure THz pulse? More importantly, how many of them are not satisfied with EOS, but rather would adopt the method proposed in this manuscript to measure THz pulse shape? EOS has been widely used in many areas of research. THz spectroscopy based on EOS is a routine measurement and many ultrafast science group have this equipment. Are they not satisfied with EOS? Will they build an accelerator to enhance the capability of their instrument? Certainly not.

In my previous comment, I mentioned that 'some measurements that can't be done with EOS should be reported in this manuscript. However, only a very ordinary measurement is reported.' By doing so, the authors could demonstrate that the demonstrated technique has wider application than EOS. However, the author's reply is 'This 'very ordinary measurement' is exactly what we want to report by elucidating its essential function as a new oscilloscope.' Well, I would take it as this method can only do ordinary measurement as EOS, but with a much more complicated and expensive method. It's more complex than necessary, if it can't measure something that is measurable by simple method such as EOS.

In summary, I can't recommend its publication in Nat. Comm, because it clearly doesn't meet the standards for innovation and impact. This work lacks a clear motivation and using a relativistic electron beam to sample the THz field limits its application to special cases.

Reviewer #3 (Remarks to the Author):

Dear authors,

the manuscript has been improved. The simulations performed on the NIR wavelength range are appreciated. They show a broad field of improvements needed to apply the technique accordingly.

Note: line 112, "Fig. 1f" is a wrong reference, it might be a reference to Fig. 1e.

Response to reviewer's comments

We would like to thank all reviewers for their efforts and constructive criticism, which allowed us to further deepen the understanding of our studies and improve the quality of our manuscript. We have taken most of the reviewer's comments to heart and revised the manuscript to clarify all the points. To each reviewer comment written in boldface, we have tried to give our response (in blue color) in detail.

● Comments of the reviewer #1:

The manuscript has been improved by adding supplementary materials to show the process of THz-electron interaction. My concern in the technical part has been largely addressed, but my concern on novelty and impact of this work, remains.

1. THz-electron interaction in a narrow slit has been experimentally studied in PRX 8, 021061 (2018) and PRAB 22, 012803 (2019) (should have been cited). Because the deflection of the electron depends on the phase of the interaction, one can either extract the information of the electron beam or the information of the THz pulse from measurement of the deflected beam distribution. The original goal of the scheme is to use the THz pulse to measure the electron pulse structure.

⇒ We added the recommended references to the introduction of our revised text. As the reviewers mentioned above, they used a narrow metal slit but the original goal of their techniques are to measure the longitudinal length of electron bunch and timing jitter/drift between electron bunches and optical pulses. These methods never capture a single-shot THz waveform in both papers due to the direction of electron-wave interaction. Moreover, L. Zhao *et al.* shows the simulation and the experimental data of the streaking deflectogram, however it does not mean the incident (original) THz waveform which was measured by EOS as shown in Fig.3(a) because they used a resonant metal slit having a finite width of 250 μm .

Page 3 in main text: "For the temporal characterization of relativistic electron bunches, electromagnetic waves have extensively contributed in two diagnostic ways: electro-optic detection with near-infrared laser pulses¹⁹ and streaking with a radio frequency (RF) wave²⁰ or a terahertz (THz) wave²¹⁻²³."

2. Roughly speaking, a small accelerator including laser, rf source, electron gun, low-level RF system, magnets, power supply, etc. costs > \$ 1 M, but the THz source costs < \$ 0.1 M. So it is a small investment to build a THz source compared to the accelerator, but it allows the measurement of important information of the electron beam not accessible by other methods. The authors of the current manuscript wants to go the other way around, e.g. using the electron beam to measure THz pulse. The investment is high, and the gain is low. As shown in the paper, EOS can do it for most of the cases, I just can't imagine who will build a small accelerator to produce electron beam for this purpose unless they already have one.

⇒ We would like to emphasize again that the ultimate goal of our study is the design and the demonstration for single-shot recording of an ultrafast light wave. The THz wave was just used as one candidate of ultrafast lights to demonstrate our idea. If not, the title of this work must be "Real-time terahertz oscilloscope...". We think that the viewpoint of "investment and gain" is not appropriate to

evaluate the novelty and impact of our work.

3. In Fig.3 the authors show the distortion from EOS by the echo signal. However, as long as one know it is an echo signal, one can remove it before FFT, which removes also most of the distortions in spectrum. EOS is a standard technique in THz spectroscopy which is widely used to study ultrafast dynamics. So the distortion is exaggerated in Fig.3. I have many friends working on THz spectroscopy, and I don't hear anyone complaining about the echo signal. In the reply, the authors said, 'We strongly point out that the rationale for the reviewer's negative judgment is not logically appropriate. As mentioned above, the inaccessibility of the ultrashort electron beam is somewhat exaggerated.' I don't agree with this statement. There is no doubt that there are >10 labs in universities that have small accelerators to produce relativistic electron beam. But the question is how many of them can produce <100 fs beam and how many of them have the need to measure THz pulse? More importantly, how many of them are not satisfied with EOS, but rather would adopt the method proposed in this manuscript to measure THz pulse shape? EOS has been widely used in many areas of research. THz spectroscopy based on EOS is a routine measurement and many ultrafast science group have this equipment. Are they not satisfied with EOS? Will they build an accelerator to enhance the capability of their instrument? Certainly not. In my previous comment, I mentioned that 'some measurements that can't be done with EOS should be reported in this manuscript. However, only a very ordinary measurement is reported.' By doing so, the authors could demonstrate that the demonstrated technique has wider application than EOS. However, the author's reply is 'This 'very ordinary measurement' is exactly what we want to report by elucidating its essential function as a new oscilloscope.' Well, I would take it as this method can only do ordinary measurement as EOS, but with a much more complicated and expensive method. It's more complex than necessary, if it can't measure something that is measurable by simple method such as EOS.

⇒ Unlike a photoconductive THz detection, EOS cannot be free from the Fresnel reflection and the phase retardation because the THz signal comes from the interaction between a THz wave and a sampling NIR wave in electro-optic crystal with a finite thickness (typically 50 μm to 1 mm). So, corresponding time-spacing of echo signals is 333 fs to 6.67 ps. Of course, EOS is a standard technique in THz spectroscopy but amplitudes of echo signals are depending on the system environment. It may be the case that some researchers remove echo signals before FFT and use the zero padding as its substitute to get more smooth spectrum. However, we feel that removal of such signals is scientifically incorrect because the removed part contains a long oscillating tail (typically up to 10~50 ps under nitrogen purging system) of main signal.

Finally, we'd like to say that our relativistic electron oscilloscope is not an alternative instrument of THz detector like EOS or photoconductive antenna. The proposed technique is a new way to visualize an ultrafast oscillation of electromagnetic wave in real-time.

In summary, I can't recommend its publication in Nat. Comm, because it clearly doesn't meet the standards for innovation and impact. This work lacks a clear motivation and using a relativistic electron beam to sample the THz field limits its application to special cases.

⇒ There is no technical way to visualize a distortion-free single-shot waveform oscillating at a high frequency (> THz) so far. The proposed oscilloscope provides its solution using a transversely-wide and longitudinally-short structure of electron bunch and a well-preserved waveform inside a metal slit.

This is the clear motivation of this study.

● **Comments of the reviewer #3:**

The manuscript has been improved. The simulations performed on the NIR wavelength range are appreciated. They show a broad field of improvements needed to apply the technique accordingly.

⇒ We really appreciate the referee's evaluation for the improvement of our revised manuscript.

Note: line 112, "Fig. 1f" is a wrong reference, it might be a reference to Fig. 1e.

⇒ We revised the wrong reference "Fig. 1f" to "Fig. 1e".